# Harmony Justesse Analogia Linguae: Literature as a "First Language of God" in the Thought of Hans Urs von Balthasar

Rodrigo Polanco

Facultad de Teología, Pontificia Universidad Católica de Chile, Santiago 7820436, Chile; rpolanco@uc.cl

**Abstract:** Balthasar is, first and foremost, a Germanist. His theology is internally related to literature. This article shows how deep this link is, starting from three aspects: the literary coordinates that appear to be integrated in the theological structure of his thought, the theological language that Balthasar discovers in the literature, and the use of some literary instruments to engage with theology. Overall, Balthasar bases this profound and internal use of literature on the harmony/harmonization (Zum-stimmen-bringen/justesse/analogia linguae) he discovers between literary form or style and theological content. This harmony, justesse, or analogia is mediated by the free creativity of the author of the literary text under the power of the Spirit and sustained by the analogia entis and Christological analogia entis.

**Keywords:** literature; Hans Urs von Balthasar; theological style; form; harmony; justesse; Theo-Drama; analogia linguae

## 1. Introduction

Hans Urs von Balthasar studied Germanistics (*Germanistik*, i.e., German literature),[1] and, despite his later theological studies, he remained a Germanist, that is, a humanist versed in literature (Lochbrunner 2007). This means that his deep connection with literature and various expressions of art—which was characteristic of him—on the one hand extended far beyond the time of his doctoral studies (Pham 2000, pp. 176–78), and, on the other hand, internally determined his theological reflection.

Balthasar never gave up literature. As a Jesuit, in his years of philosophical–theological studies, he devoted himself with enthusiasm to the reading of the Fathers of the Church and dedicated some monographs to them.[2] The philosophy and theology presented to him by some of his professors during those years seemed dull and moribund to him. Rather, he was fascinated by the great French Catholic poets and writers of the late nineteenth and early twentieth centuries.[3] Later, after his theological studies, he completely rewrote his doctoral thesis, more than tripling its size, and gave it a new title: *Apocalypse of the German soul. Toward a theory of fundamental orientations* (von Balthasar 1937–1939). There, in more than 1500 pages, he offers monographic studies on more than 20 "Germanic" philosophers, poets, and literati from the end of the Enlightenment onward. In his early years of ministry in Basel, he devoted himself to an important publishing task to maintain and transmit the humanist and Christian cultural heritage—while Europe was engaged in an inhuman, and indeed, anti-human war. From then on, he maintained constant contact, reading, study, translation, and elaboration of the anthologies and monographs of the most varied literary works.[4]

Thus, literature was incorporated naturally and internally into Balthasar's theological work in two senses. On the one hand, literature (and other forms of art and the humanities) gave him a precious instrument for engaging with theology. On the other hand—more relevant still—with that same instrument, he was able to perceive that literature itself is also a theological voice, a kind of "first language of God" (von Balthasar 1984a, p. 255). This is the object of the present article: What metaphysical and hermeneutical criteria

underlie Balthasar's theological use of literature? How is this use of literature reflected in the theological structure of his work? What is the concrete meaning of literature as God's first language? Thus, this study aims to show, in a synthetic and concrete way, how deeply Balthasar's literary training influenced the development of his theology. It is another way of understanding—in its more subtle aspects—what he said in his well-known interview: "I am by nature a Germanist [*ich bin von Haus aus Germanist*]" (Albus 1976, p. 73).

Balthasar's relationship with literature is well known and has been well studied in various aspects. There are many studies on his literary monographs, as well as, for example, theses on his Germanist formation (King 2016; Carpenter 2021, pp. 628–30). In the field of literature, however, his work as a Germanist does not seem to have had much echo (Haas 1998, pp. XXXI–XXXII; Venard 2009, p. 89). With respect to our particular point of view, we find good studies about the theological use of literature, although in a lesser quantity than those referring to his literary studies as literary works. Thus, for example, Carpenter offers "a careful exploration of von Balthasar's use of poetry and poetic language and a detailed analysis of his philosophical presuppositions". Carpenter concludes "that Hans Urs von Balthasar uses poets and poetic language to make theological arguments, because this poetic way of speaking expresses metaphysical truth without reducing one to the other" (Carpenter 2015, p. 2). J. King shows very convincingly the Germanist background of Balthasar's thought, positing that "by understanding what kind of Germanist Balthasar was, we can better understand what kind of theologian he became" (King 2016, p. 4). Ch. Denny expounds "the relationship between imaginative literature and dramatic anthropology in the first volume of Balthasar's *Theodramatik*" (Denny 2004, p. 5). Ch. Myers recently completed a thesis on the meaning and importance of the concept of "theological style", in the sense of "*how* Balthasar goes about discerning each theological style" (Myers 2023, p. 2). All these texts approach our topic from different points of view. However, none of them *explicitly* study the inner harmony (and also its outward fitness) between literature and theology as a basis (or hermeneutical criterion) for the theological use of literature in Balthasar's theological proposal: what we have called the theological voice of literature—although they indirectly alluded to its reference. Our goal is to comprehensively describe the congruence that justifies its incorporation as theological data in the elaboration of Balthasar's trilogy, the culminating work of this great author.

We will develop the examination at three moments of Balthasar's work. First, well-known but important to remember in this context are the literary coordinates that appear integrated in the structure that sustains Balthasar's theology. Second, we will analyze the theological character or language that is discovered in the literature. Finally, we will review the use of a few literary instruments to describe and understand revelation. From these three aspects or points of view, we provide insights for understanding the metaphysical–literary–theological unity that Balthasar discovers, which, at some point, he has been called theological justesse (*The Glory of the Lord III*, p. 234) or analogia linguae (*Theo-Logic II*, p. 81). This allowed him to internally incorporate the literary phenomenon into his great theological work, synthesized in his trilogy. There, he presented the Christian revelation from the perspective of *form* and theological *styles*, from the dramatic of the encounter between the divine and human freedoms, and from the word as an *expression* of being (Polanco 2021a, pp. 73–81).

## 2. Literature within the Structure of Balthasar's Theological Thought

Henri de Lubac's sentence, pronounced in homage to Balthasar's 60th birthday, is well known:

> This man is perhaps the most cultured of his time, and if there is a Christian culture anywhere, it is here! Classical antiquity, the great European literatures, the metaphysical tradition, the history of religions, the multiform essays of today's man in search of himself. And above all, sacred science, with St. Thomas, St. Bonaventure, patristics (complete), without speaking for now of the Bible . . ., there is nothing great that does not find welcome and vitality in this great spirit.

> Writers and poets, philosophers and mystics, ancient and modern, Christians of every confession, all are called to give him their note. All their voices are necessary to it, from which will result, for the greater glory of God, the Catholic symphony (de Lubac 1967, p. 184).

De Lubac knew Balthasar very well, and he testified with great accuracy to the way in which literature, in its wide range, is extraordinarily present in Balthasar's work. For Balthasar, everything "great" was "necessary" to elaborate on that "Catholic symphony", which was his lifelong claim. However, this should not be understood as a disciplinary or professional feature of his studies, but as a sign of something much deeper. It is about a way of being and thinking of Balthasar himself, which was gradually expressed throughout his life and studies. In this sense, some features of his biography (reflected by himself) are illustrative, allowing us to enter into the decisive quality of our author.

Although he was an avid reader from an early age, his early years were marked more by music. He tells us that "the most important content of [his] years before the *Gymnasium* was music. Already from the first captivating musical impressions [...] [he] spent whole hours on the piano. At the Engelberg school, he was able to participate in the orchestra for masses and also in operas" (von Balthasar 1984b, p. 31). Similarly, he informs us that:

> My youth was marked by music. My piano teacher was an old lady who had been a pupil of Clara Schumann. She introduced me to Romanticism, whose last epigones I could listen to with pleasure during my studies in Vienna: Wagner, Strauss, and especially Mahler. All this came to an end when I listened to Mozart, which I never left again to this day; even though, in my mature years, I still liked Bach and Schubert (von Balthasar 1993a, p. 420).

In fact, his first book, at the age of 20, was *Die Entwicklung der musikalischen Idee. Versuch einer Synthese der Musik* (i.e., The Development of the Musical Idea. An Attempt at a Synthesis of Music) (von Balthasar [1925] 1998). The surprisingly early date of this work not only demonstrates his extraordinary talent, but also gives a glimpse of his way of understanding reality and the arts, especially music. Indeed, he organized this "attempt at synthesis" in two parts: the constituents of music (rhythm, melody, and harmony), and a kind of philosophy of music, with which he attempted to develop an idea or theory of music. Here, his principles of polarity and organicity were apparent. M. Lochbrunner adequately asserts that, in this work, we can perceive "the author's effort on his way beyond music and philosophy toward theology" (Lochbrunner 1998, 2000, p. 326). Indeed, Balthasar concludes:

> Music is that form that brings us closer to the spirit; it is the thinnest veil that separates us from it. However, it participates in the tragic destiny of all the arts: a longing for permanence and, therefore, provisional. And precisely because it is closer to the spirit, without being able to fully grasp it, the yearning is stronger in it [...]. Music is a limit point of the human, and at that limit the divine begins. It is an eternal monument so that human beings can guess what God is: eternally simple, multiple, and dynamic flowing in himself and in the world as Logos (von Balthasar [1925] 1998, p. 57).

Balthasar understands that music brings us very intensely close to the divine and can leave us at the limit of the ineffable. Music is pierced by this duality between what is perceived and the ineffable evoked from the sensitive. Musical forms based on rhythm, melody, and harmony become an instrument for perceiving the divine, since they are totalizing, coherent, and integral forms (Guerriero 2013, p. 44). Hence, it has always been a privileged language for religious and communitarian communication with the divine. In this sense, what is most important in music (as in all art) is its human and universal dimension—that which, as a yearning, anticipates the divine that is for all, beyond its personal dimension. Balthasar concludes, "That is why we love art: it is our image; it is the reflection of our greatness and our limitation" (von Balthasar [1925] 1998, p. 54). In all this, we see, in nuce,

much of what he develops later on about the form and the "theological" quality of human expressions (Carpenter 2015, pp. 19–20).

However, at the end of his secondary studies, Balthasar turned to literature, not music. This may seem surprising, but in fact it is not if we look at it in the light of his later development. For him, starting new studies did not necessarily mean giving up music. In fact, he was always a music lover: he listened to it, played the piano, and did not fail to write some studies on the subject.[5] He continues his story:

> In Vienna, I did not study music but, above all, German literature. And what I learned there was what I later placed at the center of my theological writings: the possibility of seeing, evaluating, and interpreting a *form* [*Gestalt*], that is, the possibility of possessing a synthetic gaze (which is the opposite of Kant's critique, of the analytic, of the natural sciences). And I owe this perception of the form to Goethe, who, emerging from the chaos of the *Sturm und Drang* [i.e., storm and impetus], never ceased to see, to create, and to value living forms. I owe him this instrument, which was decisive for everything I have done since (von Balthasar 1993a, p. 420).

> I began my study of philology out of love for German poetry; I also studied some philosophy, Sanskrit, and Indo-German studies, without seriously thinking about what I would dedicate my life to. Only much later, when the lightning [*Blitz*] of vocation had already touched me many years ago, and I had finished my philosophical studies in Pullach (accompanied from afar by Erich Przywara) and the four years of theology in Lyon (inspired by Henri de Lubac) [...], I realized what a great help to the conceptual design [*Konzipierung*] of my theology the knowledge of Goethe, Hölderlin, Nietzsche, Hofmannsthal and, above all, the Fathers of the Church, to whom de Lubac had led me, would be. The fundamental postulate of my work, *The Glory of the Lord*, was the ability to see a *form* in its coherent totality. Goethe's gaze had to be applied to the phenomenon of Jesus and the convergence of the New Testament theologies. The Fathers, in their own way, had had this gaze, which retains its validity, even beyond modern analytics (von Balthasar 1984b, pp. 31–32).

Balthasar recognizes in a very significant way that his later (theological) work was marked by the "literary" concept of *form*. This is seen especially in the first part of his trilogy, *The Glory of the Lord: A Theological Aesthetics*. There, *form* is the articulating principle of all perceptions of beauty, which raptures by its splendor. It is beauty that is given in and from a concrete form. This notion allows us to understand, in a new way, the fact of revelation in Jesus of Nazareth. Indeed, the revelation of the unique and incomprehensible God is given in and through a concrete human being. It is given in a personal figure situated historically and temporally and, depending on it, also in a communitarian figure. It is given in a concrete existence, where the transcendent and universal become present, and which opens every concrete being to the horizon of the absolute (Guerriero 2013, pp. 56–57).

However, his *Theo-Drama: Theological Dramatic Theory* is also marked by this vast literary expertise. This is evident in the first volume, which he calls *Prolegomena*. There, Balthasar presents an "assembling of material and themes toward a theology", offered as scaffolding for the whole development of the drama. That is, he uses them to expose the relationship between human freedom and divine freedom carried out in the life and death of Christ. In that "assembling of material", one can see all the subsequent theological themes announced and insinuated at beforehand, "as they begin to emerge obliquely" (*Theo-Drama I*, p. 9), which Balthasar justifies:

> It seems to me that, instead of suddenly rushing into the construction of such a theology [i.e., *Theo-Drama*], one should first elaborate a "dramatic instrumentation" of the literary and lived theater, and thus of life itself, in order to prepare images and concepts with which one can then work (with an adequate transposition) (von Balthasar 1993b, p. 98).

This is one of the most novel things in Balthasar's theology: the passage from the dramatic to the theological dramatic or Theo-drama. It is a genial and characteristic way of integrating literature and theology (Denny 2004, pp. 303–5). However, it is not an extrinsic integration, less still a forced one. "It is not a question of recasting theology into a new shape previously foreign to it. Theology itself must call for this shape; it must be something implicit within it, manifested explicitly too in many places" (*Theo-Drama I*, p. 125). Indeed, it is so:

> Revelation, however, in its total shape, in large-scale and in small-scale matters, is dramatic. It is the history of an initiative on God's part for his world, the history of a struggle between God and the creature over the latter's meaning and salvation [. . .] Theology will always have to reflect on all this, without ever coming to a finished conclusion; however much it tries to create a systematic presentation, it must leave room for this dramatic aspect and find an appropriate form of thought for it (*Theo-Drama I*, pp. 125–26).

Literature was, for Balthasar, "a catalyst for theological innovation", as Lochbrunner (2005, p. 177) puts it. The topos of literature is useful not only for some of its contents. Equally useful are its structures of thought and interpretation of human life, which allow us to better understand this vast and complex phenomenon of revelation. Some examples include the drama as a mirror of one's own existence, the interrelation of the representation with the spectators, the relationship between the temporal–spatial limits of the represented work and its very broad horizon of understanding, the struggles between good and evil, and the deep integration between actor and role (von Balthasar 1993b, p. 98). All of these shed powerful light on a new understanding of revelation. To Balthasar's credit, he made it shine in his trilogy.

Finally, *Theo-Logic: Theological Logical Theory* consists of three volumes dedicated to the human word as an *expression* of a truth that surpasses it (Carpenter 2015, pp. 107–14). Here, the multiple literary forms are an expression of Being, insofar as *word*, and not mere instruments to relate facts or describe thoughts. Since creation through the Logos, reality has been constituted as an expression of the Foundation and as a place suitable for the revelation of the Absolute. Thus, "his [i.e., of God] Logos can express itself, not just vaguely and approximatively, but adequately, in the narrow vessel of human logic". In fact, "man is not just a perceiver and an actor; he is also a thinker, speaker, and formulator". The human being "attempts to translate God's logic into his own", and he does so with truth and accuracy, because the praxis of God in the incarnate Logos is also an expression *in* a flesh (*Theo-Logic I*, pp. 21–22). Therefore, it is possible to elaborate on a theory about how God acts "in human concepts and words". All such elaboration "is already anticipated in the structures of intra-worldly truth", where literature is one of its privileged forms (*Theo-Logic I*, p. 22). This is Balthasar's novelty. Consequently, all literary expression is an analogy of the expression of Being and the expression of God (Moda 1986, pp. 58–60).

The integration of literature into Balthasar's theological thought also occurs in many other moments of his work. It is permeated by several criteria that reflect its literary matrix. These are well-known issues, so we will only mention them here: the analogy of the transcendentals, as a path of revelation; the understanding of the whole in the fragment, as an interpretation of the greatness of the literary text; and the validation of beauty by its own harmony and not by external criteria, as with any true work of art; among others.

In short, literature has structural importance in Balthasar's theology. However, as expressed by Cordovilla, "the relationship between theology and literature does not arise from the fashion of the interdisciplinarity of knowledge, but from an intrinsic necessity of his work, due to the nature of divine revelation" (Cordovilla 2022, pp. 40–41).

## 3. Theological Language of Literature

1. For Balthasar, the language of the human being can also be (or is) the language of God. He dealt with this extensively in his trilogy, especially in *Theo-Logic II* (pp. 248–80). He returned to it briefly when he received the Paul VI International Award in 1984 (von Balthasar 1984a, pp. 248–76). In fact, the language of a human being possesses a double

dimension that allows it, in a second moment, to be the "first language of God" (von Balthasar 1984a, p. 255). On the one hand, corporeality and the faculty of imagination find their fullness in human beings. There, the multiple sensory impressions, unified by the imagination, are situated "in the depth of his self-reflection and in the corresponding breadth of his gaze on the world in its totality" (von Balthasar 1984a, p. 251). He looks at everything from his "heart". Thus, "the human being speaks with his whole psycho-physical unitary being, and often speaks more truthfully from the expressions of his face and his gestures, conscious or unconscious, than with his words" (von Balthasar 1984a, p. 252). This is the way in which human language develops all its expressive power, which is then embodied, for example, in narrative, poetic, or dramatic literature. In this marvelous capacity to express and express oneself, the human being encounters a "second dimension", "opposed to the first". It is the "experience of his own contingency: it was not he who put himself into existence here [*Dasein*], nor he who gave himself to exist as this kind (or degree) of being [*quidditas*]". For that reason, he does not have the capacity to interpret "from himself the ultimate meaning of his own form", even if he can express himself freely. He has been donated and owes it to another, who puts him where he is and how he is. In this way, he finds himself in tension with a double reality: his existence is due to others (i.e., his parents and the surrounding reality, etc.). At the same time, he is not a simple product of nature, but there is in him a subjectivity that projects him toward an absolute subjectivity, which gives reason for his existence (von Balthasar 1984a, pp. 253–54). Now, every human being attempts a synthesis:

> The synthesis between the positive that is to exist in one's own essence (always limited and shaped) and the positive that is to exist in the fullness of Being, beyond all limits. This synthesis is absolutely necessary to be able to open oneself to the ultimate reality, the giver of meaning. And yet, it is also absolutely denied to him, because he recognizes it only within his own finitude (von Balthasar 1984a, p. 254).

This openness to the absolute, which is given on the basis of one's own finitude, in no way allows one to know what God is, because it is anchored in one's own finitude. However, it does allow it "to be able to understand what God is not", following St. Augustine (von Balthasar 1984a, p. 255).[6] In different ways and to different degrees, the human being recognizes the following—although he can also deny it:

> All finite beings are permeated by *logoi* by unity and fragments of meaning. In the human being, these *logoi* are found to maximum degree. These recognizable fragments must find their origin in unitary *logos*, for which, however, human discourse and reasoning can no longer find any word (von Balthasar 1984a, p. 255).

This is where the theological voice of literature emerges. In it (literature), we also find those multiple "fragments of the *Logos*, scattered in finite and created things" (von Balthasar 1984a, p. 255). They emit "a sonorous language", which, at the same time, is "wordless", because those who speak are those created logoi, not the Logos itself. Indeed, "they speak positively of him [i.e., *Logos*], proclaiming that he exists, because he has given them existence. They speak with silence because it is not he [i.e., *Logos*] who speaks". However, one can speak positively of God because there is a proximity (not spatial, but of Being), since all the beings of the world express themselves and express the Being that sustains them. This they do from their form, in which their beautiful interiority shines forth. Thus, although God speaks from silence, "it cannot be denied that every worldly entity, insofar as it expresses itself, is also an expression of Him. Without it being possible to say that any of them is its own expression". Therefore, all speech of the creature, which derives from God, speaks *about* God and, at the same time, is a silence *of* God—which is a sonorous silence, according to the expression of Ignatius of Antioch (von Balthasar 1984a, pp. 255–56).[7]

Another aspect is noteworthy here. Since those created logoi derive from God, then God himself (as God) can speak authentically from that created word with an explicit

revelation in the flesh. This "authentic speaking of God maintains a background of mystery, which obviously does not mean that the Word lags behind what is said, but that it announces itself in the (divine) dimensions of the Word" (von Balthasar 1984a, p. 257). For this reason, word and silence always remain united, and the believer can "listen" both to the mystery (i.e., silence) hidden in the depths of the revealed Word, and to the word spoken in the silence (i.e., mystery) of creatures, especially human beings.

> From the Christian perspective, all this reveals a dimension of the original language of God that manifests itself in creation as such. Everything that expresses itself in the world speaks, referring to one who has granted it the faculty of expressing something of himself [i.e., God] (beyond himself), without him [i.e., God] having had the need to introduce himself into the alienation of expression (von Balthasar 1984a, p. 257).

In synthesis, from this anthropological and metaphysical foundation of human language (and of revelation), Balthasar developed a profound theology in a particularly creative way. In a very accentuated way, he integrated that particular genre of this "first language of God", which is *literature*, a genre that is also very expressive. It has the capacity to show Theo-drama in action—that is, the rapturous encounter with beauty, the struggle for good, and the intuition of truth. It can do so in all its artistic and structural dimensions (Moda 1986, p. 51), starting from a particular story that is unrepeatable and concretely situated in time and history. Balthasar focused on the singular events of history and narrative because they claim us as a divine voice (Johnson 2011, p. 84).

This theological reading of literature is exemplified in some works of Paul Claudel. Balthasar, who was a friend and whom he admired, comments on this in the postface to Claudel's masterpiece, *The Satin Slipper* [*Le soulier de satin*]. Balthasar translated it several times until he was completely satisfied with his work (von Balthasar 2005, p. 162). In the work, he affirms that, faced with sublime works of art, such as the *Divine Comedy* or *Faust*, the best way is to go through them without previous schemes. One must let oneself be led by the work itself, as it pleases, so that it reveals itself. Thus, for Balthasar, Claudel's literary work has a profound theological horizon that permeates the whole plot: "How is it possible to belong, at the same time, completely to the world and completely to God?" (von Balthasar 1939, p. 377).

The play begins with a holocaust or prayer of a Jesuit priest who, dying, offers his life for his brother Rodrigo so that he may find God. The priest asks God that if his brother does not find him through a clear path, he should find him through a dark one. That is the leitmotiv of the work, expressed in the two aphorisms that introduce the text: *Deus escreve direito per linhas tortas* [*God writes straight on crooked lines*] (Portuguese saying); *Etiam peccata* [*Even sin*] (Augustine). The plot unfolds in the existence of the protagonists, day by day, as an adventure. There, "the necessity and impossibility of Rodrigo and Prouhèze's love are a parable of the same necessity and impossibility of existence" (von Balthasar 1939, p. 379). However, existence does not allow itself to be reduced to the apparent, simultaneously promising fulfillment. Prouhèze intuits it by leaving her satin slipper to the Virgin, intertwining her being completely of the world with her being completely of God.

Thus, the drama, full of literary resources where irony is not lacking, develops from the relationship between love and death, a classic place in literature since the Greek tragedy. "Love can reach such a strength and definitiveness that makes it rival death" (*Theo-Drama I*, p. 374). Here, love is transfigured by death, either suddenly (Prouhèze) or slowly, accompanied by stripping and humiliation (Rodrigo). However, the drama also develops the relationship between love and the world. For Balthasar, in Rodrigo and Prouhèze, his earthly passion is a religious theme, even though it is also pierced by the very human reality of sin. Every existence has lights and shadows; it has things that are understandable and others that are insoluble for our comprehension, because that is simply the reality of the world. In that same reality, love is transformed into sacrifice, into offering. It is even transformed into its own sword that pierces the other beloved, precisely so that this love may be transfigured into an eternal love, in the manner of the cross. Further, this love

is what guides Rodrigo to conquer the world and integrate it around his unity–totality. Finally, we find that the destiny of each of the lovers is linked to the destiny of humanity. In this sense, Prouhèze, in a dramatic and totally human way, will accept being a "star" for Rodrigo. Hence, they have to renounce even their own love for the task to which they feel called. Thus, the path undertaken (their own existence and their love) is, finally, a path of purification, a path of stripping, a path of humiliation, and a path of offering. However, it is a path that, far from separating them, brings them closer together, especially because it unites them around the definitive.

Here, the literary text is 'transformed' into a revelation of the mystery of the human being and his life (i.e., vocation) on earth. As B. Quash says, all this is possible because "every moment of time, and of the action of human agents in time" has eternal value. We must recognize that, for Balthasar, "the fact that finitude and non-repeatable particularity can genuinely mediate divine purpose. Or, put another way, finite persons and possibilities can be bearers of glory" (Quash 2006, p. 406). J. King resolves "that a great deal of what was most distinctive about Balthasar's theology—some of which is often viewed as idiosyncratic or arbitrary—was the result of Balthasar's 'theologization' of New Germanistics". This meant studying the text in a "bold, *geistesgeschichtlich* [i.e., as an intellectual history], and synthetic" way (King 2016, pp. 298 and 302, respectively).[8]

In another literary context of Balthasar, A. Haas called all the foregoing notions "a true theological a priori, which permeated all his Germanistic work or, in general, his literary-philological work" (Haas 1998, p. XXVII). This had already become clear in his doctoral dissertation (von Balthasar 1930), in which he worked on this theological a priori from an eschatological point of view as an unhiding of the human attitude toward ultimate, transcendent realities. The (German) literature, from its longings and proposals, and from its multiple literary genres, gives glimpses of created logoi that point to the Logos. However, it often does so from what appears to be its opposite. Dedicated to theology, Balthasar consistently maintained this "broadening of the Christian gaze open towards culture" (Haas 1998, p. XXXI). Thus, "what Hans Urs von Balthasar has repeatedly attempted, implicitly or explicitly, in his theologically oriented studies, can rightly and justly be called the theology of literature" (Haas 1998, p. XXXI). We can even speak of literature as "*locus theologicus* [i.e., source of theological knowledge]" (Haas 1998, p. XXXI). It was always a matter of understanding literature from its "religious dimension;" thus, he was able to place it also within his theological construction, because he "recognizes the hidden presence of God in every form of the world" (Kuschel 1992, pp. 115–16; Carpenter 2015, p. 158).

2. Balthasar discovers and extracts the theological background of literature with the help of some theological and literary criteria (or hermeneutical principles). Some of them have become characteristic of his entire theological proposal. We will mention three that are especially significant.

The first is *integration*, understood as "the Whole in the fragment", as the title of one of his books states (von Balthasar 1990; Avenatti de Palumbo 2012). This is another way of understanding the *Gestalt/form*, and it orients us to Balthasar's way of interpreting texts and their content. Integration "is the spontaneous art of always aiming at the Whole through the fragments of truth discussed and lived. The whole, then, is always greater than us and our powers of expression, but precisely because it is greater, it animates our Christian life" (von Balthasar 1993b, p. 105). Indeed, it is quite true that, in every beauty "to be encountered within the world", we grasp "only a fragment and an aspect of the total object" (*The Glory of the Lord V*, p. 598), but this does not prevent that fragment from containing some truth. This is so because "the aesthetic experience is the union of the greatest possible concreteness of the individual form and the greatest possible universality of its meaning or of the epiphany within it of the mystery of Being" (*The Glory of the Lord I*, p. 228). All this is possible for two reasons. First, this is because all reality (and, in particular, the human being) has been created "in Christ", according to the divine Archetype, in such a way that "all that is true in the world 'hold[s] together' in him (Col 1:17)" (*Epilogue*, p. 89). Therefore, everything that exists, concretely manifested in its transcendentals, is anchored in the beautiful, good, true,

absolute, unique, and transcendent. It reflects them in its existence. Second, by bringing the above to its fullness, once and for all and in a totally ineffable way, the Whole in Jesus of Nazareth becomes flesh and, therefore, lives and acts *as a* fragment. Consequently, from then on, the fragments can "be places *in* which the Whole manifests itself" (Avenatti de Palumbo 2012, p. 168). The Whole (the Son incarnated as Archetype) is always in each fragment of reality, and, therefore, each fragment can and must be understood from that Whole.[9] Only in this way can it be understood in its deepest reality. This has been the irruption of the vertical into horizontal history, which then unfolds as a Theo-drama: a common history of God and human beings that unfolds from that ineffable relationship between infinite freedom and finite freedoms.

In these circumstances, literature (drama) "transforms the event into a picture that can be seen and thus expands aesthetics into something new (and yet continuous with itself), while at the same time it is already translating this picture into speech" (*Theo-Drama I*, p. 21). Literature is thus a concrete form of human expression that seeks to say the original in a decidedly concrete form: a story, a poem, a figure, or a representation, etc. In that same story, the Absolute can also speak, not only in an understandable way, but in a truly transparent way, although always after the effort of translation and openness to the interiority of that word and without ever ceasing to be a word that "remains silent" (Avenatti de Palumbo 2004, p. 53). In concrete terms, this means that the voice of God is in the concrete story, but it is not exactly the story. It is necessary to know how to find behind the story the mysterious Word of God, spoken in a language that only the "heart" understands, because it is truly an *encounter* with God.

A second hermeneutical principle is what C. Avenatti calls "the principle of *otherness*". Literature, as the manifestation of a beautiful form that impacts us and gives itself to us with its word, "is the singular emergence of *that other* that bursts in and that the subject cannot dominate" (Avenatti de Palumbo 2002a, p. 3). Indeed, every literary work, as a manifestation of the transcendentals of Being, is a *form* that expands into a *drama* (i.e., action). The text impacts with its beauty, speaks with its truth, and makes the reader discover an "internal logos, waiting to be unveiled", "which dramatically challenges man". In this concrete form, the truth shines "as the very foundation of the aesthetic form that "shows" and "gives" itself, "saying" in a concrete and delimited form the inexhaustible richness of Being as good truth" (Avenatti de Palumbo 2002a, p. 9). This form is an action or drama—that is, a relationship of freedoms, historically pierced by the verticality of the gratuitous presence of God. Thus, this form radiates and impacts with its beauty. Further, being an action of liberties, it is also equally a drama moved by love that gives itself. In other words, it is a form that "'exists' in the dynamism of self-giving" (Avenatti de Palumbo 2005, p. 29).

The reader is thus introduced to that "reciprocity between benevolence and thanks-giving, between grace and gratitude", which speaks and challenges from the core of the drama as an "'admirable exchange' of love" between infinite freedom and finite freedoms (Avenatti de Palumbo 2005, p. 29). In this way, the text challenges finite freedom itself and invites us to understand our own existence as "'gratuitousness' offered as a gift" (Avenatti de Palumbo 2005, p. 28). Here, the reader is confronted with that profound and paradoxical human reality, which is, on the one hand, his finite, historical, and temporally delimited existence. On the other hand, this existence is in an ineluctable relationship with infinite freedom, expressed in multiple forms: from destiny (in antiquity) to providence (in Christianity). In this confrontation, the human being enters into a hidden relationship with this infinite freedom. Thus, this literary dramatic form (whatever its concrete form) appears as a "free gift", one that is very similar to what has been theologically called "grace" and of which it can become an expression. In Balthasar's words, "Here the artifact becomes transparent, allowing the absolute to shine through it "in a mirror dimly", revealing permanent, abiding value in something that is unique and of the moment, irrespective of the changes of style" (*Theo-Drama IV*, p. 103). This is the paradoxical task of the poet. According to

Avenatti, his task is "to say the absolute in temporary time, to insert the definitive into the relative transitory by means of the fragile word" (Avenatti de Palumbo 2005, p. 26).

This is where literature becomes theology, if we understand theology in its deepest sense, as *Theo*-logy. We have said that the various literary forms have a mirror function: "to be a place where man can look in a mirror in order to recollect himself and remember who he is" (*Theo-Drama I*, p. 86). This means that they are an *other* in front of us, which challenges us with its reflection. Thus, each literary work, in all its autonomy and its otherness, is impregnated with referentiality (Avenatti de Palumbo 2002b, p. 24). Therefore, whoever encounters literature is confronted with an other who speaks metaphorically of himself. In these others, he encounters very deeply the *Other*, who challenges him in an even more profound, existential, total, and permanent way.

We know that, deep down, and from our point of view of faith, this reflects the Trinitarian dynamic imprinted in creation. Indeed, from all eternity, the Father and the Son reflect each other, and their very being (or hypostasis) consists precisely of their referentiality to the other: they are "subsistent relations", Thomas would say (S Th I q29 a4). Thus, every human being, created in the image of the Son, also lives from that Other through the Son in the Holy Spirit. In this way, every created other (even more so a beautiful form) will always reflect the Other/Son, who will help us to encounter the Other/Father, in whom "we live and move and have our being" (Acts 17:28). This is the profound theological dimension of literature in so far as it is beautiful, good, and true.

Third, there is the principle of fundamental *polarity* or paradox that runs through all created reality. This is one of the fundamental theological principles that structures Balthasar's theology. Indeed, starting from the "real distinction" (of which Thomas speaks), which encompasses the fact that we exist but that we could not exist—every created being is open to the Absolute (von Balthasar 1993b, p. 112). From here is born that insurmountable paradox that frames everything created: the transcendent Absolute is the only one that gives true consistency to contingent being, but it does so in such a way that it always maintains that Creator–creature distance, which is insurmountable. That is the fundamental polarity of created being: all being is traversed by a constitutive and basic duality, which implies a constant tension between the poles. At the same time, it is a tension that makes possible the development of its own being. Thus, we experience that Being only appears in a form, but that form refers to Being in its totality; every being needs others, but they can only be given freely; and, finally, every being expresses its interiority, but does so only from an articulation of previously arranged gestures or language.

That paradoxical polarity also permeates revelation in its basic articulation. If God is indeed revealing himself through his words and actions, then he is necessarily showing something of his own nature as God. However, in revealing himself, he also reveals his incomprehensibility. In the plenitude of time, the incarnate Son is the summit of revelation because he is the authentic manifestation of the Father: he unites in himself—the visible and the invisible—in a completely paradoxical simultaneity. "Only now does the full paradox become visible. For if God's Word has become flesh, then everything that is to be disclosed—despite every seeming impossibility—must become present in this 'flesh,' in this finite and transitory existence. It is from it that everything must be drawn forth" (*The Glory of the Lord VII*, p. 143).

For Balthasar, this paradox also transcends literature and thus gives it its profound "theological" condition. In fact, he states this, albeit in reference to a specific literary form, but applicable to all literature:

> Existence has a need to see itself mirrored (*speculari*), and this makes the theatre a legitimate instrument in the pursuit of self-knowledge and the elucidation of Being—an instrument, moreover, that points beyond itself. As a mirror it enables existence to attain ultimate (theological) understanding of itself; but also, like a mirror, it must eventually take second place [...] to make room for the truth, which it reflects only indirectly (*Theo-Drama I*, pp. 86–87).

This "theological" capacity of literature, which "indirectly" makes the truth of being (and its foundation) shine forth, has its origin in creation, which has been realized under the archetype of Christ (Col 1:17). Thus, when the Logos assumes the flesh, not only does he manifest himself in that flesh, but he also illumines *in* the flesh (i.e., in everything created) its Christological foundation. Then, illuminated by revelation, creation itself shines forth (in itself and with a new light) the Foundation that created it and that constantly sustains it. In other words, the created receives a new light to reveal, mysteriously and paradoxically, the Absolute (*Theo-Logic I*, p. 13). In this way, literature (dramatic, poetic, or narrative), the sublime expression of created humanity, can also be read and understood as a true expression of the Absolute and an authentic gift from the original source of all that exists. Thus, literature is traversed by that essential paradox "which consists in the fact that the infinitude of life can be expressed and find its consummation in the letter, rhythm and method; in spite of being spirit and form, ineffability and immeasurability" (Avenatti de Palumbo 2007, p. 282). Literature, in its multiple forms, poses "the question of the absolute horizon of existence". But it does so not only as a "metaphor", but by referring to the "quintessence of life itself", since "as a human production it is religiously impregnated", as Hass states (2000, pp. 19, 20, and 27, respectively). This is an anthropological constant that runs through the whole of human existence: every literary work refers beyond itself and opens the horizon of salvation, in order to understand humanity in a new way—hence, its *theological* character, even its revealed character. Consequently, "literature realizes in each of its expressions a model of existence that represents an 'ultimate instance,'" recalling Balthasar's first intuition in his doctoral thesis (Haas 2000, p. 27; see ibid., p. 23).

In short, based on these three hermeneutical principles, Balthasar makes "a deconstructive reading" of literature in the true sense of the term. Haas explains, "Its negative element consists in the disappropriation and de-fictionalization of metaphorical allusion. On the contrary, its positive moment develops in the construction of a context of event and action under the artistic direction of God [*unter der Regie Gottes*]" (Haas 2000, p. 29).

## 4. Use of Literary Instruments to Conduct Theology

We have said that the structure of the trilogy has been marked by Balthasar's literary background. We have also emphasized how literature possesses a theological substratum that must be discovered. These notions can be better perceived in their depth and structure if we go through some moments of this great work. We will offer only a few examples of the use of literary instruments in his theological development.

Balthasar's *A Theological Aesthetics* is a theology elaborated from transcendental *beauty* and the romantic concept of *form* (King 2018, pp. 144–47). Through it, the glory of the Lord—that is, his divinity, which is proper and specific to God—is manifested in an absolutely free and unnecessary way. Therefore, in this work, Balthasar reviews the history of metaphysics, exposing what he calls *theological styles*. If we understand metaphysics "in its most original and broadest range", that is, as knowledge of the origin of things and as the study of what is true, good, and beautiful, a theology is not possible "without constant reflection on the subject of metaphysics" (*The Glory of the Lord IV*, p. 12). Then again, the unavoidable starting point for the realization of a theological aesthetic of these characteristics is "the unity between the decision to believe in the divine revelation in Christ and the universal affirmation of all metaphysical-religious truth, which is made possible and indeed obligatory in the Christian realm". This unity "is itself unique" (*The Glory of the Lord IV*, p. 15). The center of God's revelation is grasped only by the one who does not place its ultimate meaning in the cosmos, nor in man, but in God himself, and he does so as a response of love to a first love. This response must integrate faith together with "that other universality, the universality of the human spirit, already to hand in the world from the fact of Creation, the spirit, which of its nature is open to understand the being of all that is" (*The Glory of the Lord IV*, p. 11). A biblical concept without analogy in creation would have no meaning for the human being. This concept can be "revelation", "faith", "love",

"perception", or any other. God cannot act outside the transcendental laws of creation, even if his sovereignly free revelation cannot be anticipated.

Balthasar sees this integration between creation and revelation in the theological styles of a wide range of authors[10] when they are faithful to both poles and respect the asymmetry of this polarity. Our author brilliantly discovers a decisive correspondence between the glory of divine revelation and the form through which it is expressed—that is, a correspondence between theology and literary form in the deepest and most unitary sense of a *theological* form. He affirms:

> The formal object of this investigation [i.e., clerical and lay styles] is the glory of the divine revelation itself, in the multiplicity of its manifestations and understandings, and then, certainly, within that glory theological beauty as such, in its transcendence over all models of secular beauty. Thus, the object as such does not embrace directly the choice of secular means of aesthetic expression by individual theologians, as they need them for the presentation of their vision (*The Glory of the Lord II*, p. 22).

What is sought after and of interest are God himself, his word, and his grace. Everything else, such as "the secular resources of style, as they are utilized for human utterance in poetry and prose, in rhetoric and didactic, and are available for use by the theologian" (*The Glory of the Lord II*, p. 24), "has value and meaning only in so far as God is expressed and presented therein, in so far as it is transparent to God and returns to God" (*The Glory of the Lord II*, p. 23). In this way, the Word of God becomes present in human speech, starting from a double mediation: "the general phenomenon of the freedom of human expression in spiritual utterance and the humanity of the historical revelation of salvation" (*The Glory of the Lord II*, p. 26). In fact, with respect to the first mediation, in every work of musical, plastic, or literary art, as well as in theological work, "the will to express itself not only freely creates suitable form; it incarnates in this very form its freedom" (*The Glory of the Lord II*, p. 26). This freedom embodied in the literary form of theological expression is what gives this expression all its theological depth, and also "gives to the form the radiance from the depths" (*The Glory of the Lord II*, p. 26). Regarding transcendental beauty (that which can reveal the glory of God), we are not referring to some external harmony or proportion that is the fruit of human will (Morrow 2004, p. 185). Rather, we refer, in the first place, to that "considered freedom which is manifested in it and is 'necessitated'" (*The Glory of the Lord II*, p. 27), and that identifies itself with the form and is reflected from it. This is the theological *style*. For Balthasar, the core of this concept is the linkage of style with charisma, accurately described by Myers. Theological style is "the glory of divine revelation brought to expression by the freedom of the theologian who is guided by the form of revelation, the teaching of the Church, and the internal pull of charism". This is how "the particular work of the Spirit [is manifested] in the theologian's own life and work" (Myers 2023, p. 57).

With respect to the second mediation (the human nature of divine revelation), there exists an analogous (although obviously not the same) manifestation. In the human mediation of revelation, all possible literary forms are used so that divine sovereign freedom can express divine glory with the maximum of its possibilities and full freedom. It enacts this without being chained to the chosen form, which can range from whispering to exhortation and from words of consolation to chastisement. Now, the Incarnation is the "high point" of revelation, which acknowledges and sanctions "the created vessels of expression". In an unsurpassable way, we find "a total harmonisation [*Zum-stimmen-bringen*] of content and form, and this precisely in the making manifest of the divine freedom". Here, we come to the core of Balthasar's aesthetic proposal. "The phenomenon of revelation is only truly encountered by those who [. . .] see the greatest freedom of the manifestation in the greatest necessity of the form of manifestation" (*The Glory of the Lord II*, p. 27). This is what determines the harmonizing center of "style", and this is an aesthetic concept.

Now, this *harmony* is between the three poles. The divine freedom (1) expresses a content that is identified with God himself. This content is necessarily expressed in a concrete worldly form (2)—a form that reflects divine glory. However, this worldly expressive form also depends on free human creative power (3). In that expressive form, human freedom is amalgamated with the unique strength and richness of the exceptional content it seeks to express. Thus, theology, as a *style*, is a human expression (beautiful) of the divine expression (glory),[11] although on the condition that there is an authentic harmony between both moments: "obedient repetition of the expression of revelation imprinted on the believer" and "free sharing in the bringing-to-expression in the Holy Spirit [...] of the mystery which expresses itself" (*The Glory of the Lord II*, p. 28). This is an authentic theological style in which form and content, theology, and literary form are no longer separable. "Only beautiful theology, that is, only theology which, grasped by the glory of God, is able itself to transmit its rays, has the chance of making any impact in human history by conviction and transformation" (*The Glory of the Lord II*, p. 15). Only this can truly be called theology.

There are undeniable romantic undertones to these statements that run through all of Balthasar's theology. They are aesthetic–literary concepts found in the two volumes of Theological Styles (but also beyond them) where this harmony is concretely seen. Balthasar even shows that the same twelve authors he presents in these two volumes describe this harmony with their own concepts. For example, Anselm speaks of "convenience" (Konvenienz) (*The Glory of the Lord II*, p. 27), and Pascal speaks of concordance or rightness (justesse) (*The Glory of the Lord III*, pp. 233–38). These are aesthetic concepts, and, in Balthasar's interpretation, they must be understood as such. Obviously, this is neither the time nor the place to discuss whether Balthasar interprets them properly, or whether Balthasar's theological aesthetics are acceptable in all their aspects ([Brown 2018](#), pp. 180–85). I mention these concepts here only to show some examples of the use of literary instruments in the development of his theological aesthetics.

Balthasar's *Theo-Drama* begins with a large volume of *Prolegomena*, in which he presents "the whole phenomenon of theatre" (*Theo-Drama I*, p. 9). He analyzes its structure, its processes, and all its dramatic instruments, with the aim of better understanding the action of God together with the human being in the world. We have said that this is because the human being "mirrors" himself in the theater, which offers him a suitable instrument for his theological self-understanding (*Theo-Drama I*, p. 86). In this way, Balthasar spends long pages exposing and explaining "dramatic resources" (*Theo-Drama I*, pp. 135–478). He analyzes how "the idea of the 'world stage'" has been present throughout history, and then exposes the various "elements of the dramatic". These include, on the one hand, the three elements of dramatic creativity and those of the dramatic realization, and, on the other hand, how drama illuminates existence, in its tensions of finitude and transcendence, and of the struggle between good and evil. It culminates with the theme of "role" or "function". Here, the role, as a function of the "theological" mirror of the human being, is much more than "like a random dress and can be replaced by another at any time" (*Theodramatik I*, p. 604 [*Theo-Drama I*, p. 645]).[12] More deeply, it interprets "that dualism—which everyone experiences, whether fleetingly or constantly, superficially or profoundly—between what I represent and what I am in reality" (*Theo-Drama I*, p. 481). However, this culmination is only a transition. It is now a question of passing from the role to the mission or vocation—the true end of the journey—since it is of a theological order. This means responding to what God has prepared for the human being and where he wants to lead him. It is the final answer to the initial question of each person: Who am I?

Thus, after the prolegomena, Balthasar developed the Theo-drama in four volumes. Starting from the relationship between finite (human) freedom and infinite (divine) freedom, the core of the whole drama is the action of Christ among men as a Trinitarian, salvific, and eschatological action. Contrary to what might be thought at first glance, in these volumes, the dramatic resources to which he dedicated his long prolegomena appear adequately integrated, although this is in a very subtle and almost imperceptible way for a

non-attentive reader. Of course, he did not do it in a crude way, as if he were attempting formal and external parallels. On the contrary, the articulation is internal. It is a matter of making the global form created by the author signify "the whole of reality in microcosm, and it is to this reality that the author wishes to direct his audience's attention". In this way, this global form can be called "poetic justice [*justesse*]", indicating that it points "to the unattainable metaphysical justice" (*Theo-Drama I*, p. 279). The dramatic resources are revealed as a grammar of life and theology.

Thus, this literary theme of "function" or "role", which is at the center of the prolegomena, appears in the following volumes and is transformed into "mission". It is identified with the "person" of Jesus Christ, to whom we all join with our own "vocation", which also reveals to us the ultimate mystery of our existence. The identification of person and mission in reference to Christ is at the center of the entire *Theo-Drama*. In fact, every person who really takes his mission or vocation seriously, in some way ends up identifying with it. This is even more the case with Jesus in an absolute and ineffable way. He is the one "sent" (Lk 4:43); his life *is* his mission. He exists *to* proclaim that the kingdom of God is at hand (Lk 4:16–22) (*Theo-Drama III*, pp. 154–63). Thus, in him, person and mission are identified, just as missions are identified with intratrinitarian processions (Polanco 2021b, pp. 77–79). The same can be said—albeit analogously—of the vocation of every human person, even though now in a historical, contingent, and limited way. The vocation of every human being defines what each one becomes in Christ, because it is what God wants for him, what God thinks of him, and what God gives him to become. All of this deserves a more detailed study, which we cannot perform here (*Theo-Drama III*, pp. 149–202; McIntosh 2004). Suffice it to highlight the inner harmony that Balthasar finds in the mutual relationship of these concepts, one literary/dramatic and the other purely theological. The dramatic concept allows for a better understanding of the revealed theological reality (Denny 2004, pp. 11–12).

The structure of *Theo-Drama* also contains other elements of dramatic resources. As discussed earlier, the use of the idea of the "Theatre of the World" gives Balthasar occasion for "a religious and ultimately theological interpretation of existence" (*Theo-Drama I*, p. 249). There is also a series of leitmotifs and some structures of representation that make up dramatic literature, and which are also leitmotifs proper to human existence. For Balthasar, these existential leitmotifs and dramatic structures are also articulating elements of *Theo-Drama* itself. Let us look at two examples of the use of leitmotifs. (1) "The distinction between the (temporal-spatial) finitude of the performed play and its nonfinite meaning" (*Theo-Drama I*, p. 250) warns the spectator that he is under transcendent judgment and that every finite action can have eternal value. This leitmotif structures human life and gives all its theological–anthropological depth to the fact that, in this concrete man, Jesus of Nazareth, the eternal Word of God can personally act. The Word, with all his supra-temporality and transcendence, can be present within a history totally subjected to time and space. The limited and finite can become a real vehicle of the infinite. (2) "The distinction between the actor's responsibility for his performance and his responsibility to a director" (*Theo-Drama I*, p. 254) shows the deep intertwining between the profound freedom of the actor to execute his role, the fidelity to the role assigned to him, the performance of the other actors who interact with him, and the direction of the director who articulates all these factors as a whole. All this vitally interprets the work as it was conceived. On this whole, the actor is in a vital and creative tension that allows him to freely invent his own way of being the assigned character. This leitmotif, typical of all representation, appears repeatedly underlined in the Christology and anthropology of *Theo-Drama II* and *III*. Jesus freely develops and creates the way of being the Messiah (*Theo-Drama III*, pp. 198–99) (acting) from his consciousness of being the Son and sent in the face of all the vicissitudes of history and human responses (the other actors). He achieves this under the guidance of the Spirit (director) in order to be faithful to the will of the Father and to the mission entrusted to him (work). The same is true—analogically—for human vocation. Each person receives a vocation that gives him his unique and unrepeatable character and constitutes

him as a person. However, he must develop it freely and historically. The human person gradually realizes and identifies himself with that vocation/mission, in order to *become* what he is called to be, but which, at the same time, in some way, he is already (*Theo-Drama III*, pp. 207–8).

Regarding dramatic structures, we have just exposed, in an indirect way, the trinitarian reading that can be found in the triad of author, actor, and director. However, the three elements of dramatic realization are assumed throughout Theo-drama: presentation, audience, and horizon (*Theo-Drama I*, pp. 305–23). This triad refers to the global horizon with which the spectator is confronted in the performance. The audience arrives with its own expectations, which bind it to that totality of meaning or horizon "horizon within which the dramatic action takes place" (*Theo-Drama I*, p. 314), and forwards it to the ultimate goal that the author has intended, with which every spectator must ultimately confront. This structure highlights some important contours of the whole economy of salvation that are otherwise less visible. The life of Christ is a re-presentation, that is, it is a life for others, a pro-existence, to be welcomed by the audience, a welcome that is nothing other than faith. However, spectators cannot remain indifferent when they enter into a relationship with the global meaning of the dramatic action, which is now a *Theo*-dramatic. There is a communion of expectations that makes them respond with their *yes* or *no* to the interpellation of the drama. Balthasar later summarizes this in a concentrated and compelling response by affirming that history is under the sign of the Apocalypse (Lindle 2022, pp. 53–84). He affirms that history is in a dramatic form and motif: it is the necessary taking of the position of every human being in the face of the fact that the ever-greater commitment of God is awakening an ever-greater opposition to him (*Theo-Drama*, IV, pp. 45–57). Here, the relationship between expectations, the horizon of meaning, and presentation sheds new light on the relationship between the proclamation of salvation and the believing response.

We must say that the theological examples referred to here have yet to be explained in their foundations and nuances, which we cannot undertake here (Quash 2004). However, if we accept them as valid, even provisionally, then it is evident that the literary-dramatic structure marks *Theo-Drama* much more deeply than appears at first sight. Its foundation is again in theological and metaphysical harmony, of which literature is its expression. This is precisely what he will further develop explicitly in *Theo-Logic*, although he has already assumed it implicitly, both in *Theological Aesthetics* and in *Theo-Drama*, due to the mutual immanence of the transcendentals of being (*Theo-Logic I*, p. 10).

We said that *Theo-Logic*, in reviewing the transcendental *truth*, was an explanation of how to translate God's logic into human logic in a truthful and just way. The answer must be clearly Christological. Jesus, the Christ, is the concrete analogia entis, the definitive union of God and human beings (*Theo-Drama III*, p. 221). Indeed, "the incarnate Logos has succeeded in validly translating the logic of his divinity into the logic of his humanity", but only on the condition that the "Spirit of truth" accompanies this process, introducing us "into this truth of Jesus", which is the right exposition of God (Jn 14:6: I am the truth). Only in this way is the human being introduced "into the rightness of the logic of the Logos" (*Theo-Logic I*, p. 18). We know that this is possible because the "worldly being as a whole" has been constituted as a divine image, due to the analogia entis that unfolds in an analogy of the transcendentals of Being. This occurs from the very creation of human beings. However, the presence of the transcendentals of Being cannot be understood as certain "finite content". "The transcendentals, by contrast, are all-pervasive and, therefore, mutually immanent qualities of being as such" (*Theo-Logic I*, p. 15): beautiful, good, and true. This is why a narrative description shows this transposition of logics better than an abstract definition, since it does justice to the depth of the divine likeness in the human.

In addition, this Christological response is a response to the radical invisibility of the Father. "No one has ever seen God" (Jn 1:18). "The invisibility of God is firmly established in the Old Testament. It is not rescinded in the New" (*Theo-Logic II*, p. 66). The essential ontological difference between God and the creature "is 'sublated' in Christ in both senses of the Hegelian term. Christ, going beyond the law and the prophets, presents himself as

the expositor of God as he is in himself. Indeed, he unveils in his own visibility the invisible God, even while simultaneously leaving this God his fatherly invisibility" (*Theo-Logic II*, p. 67). "Whoever has seen me has seen the Father" (Jn 14:9). Here, the "fundamental law" of transcendental truth is maintained: the "mystery is not . . . something 'beyond' the truth, but . . . a permanent, immanent property of it" (*Theo-Logic II*, p. 68). Thus, revelation, although it unveils the invisible God (the truth is known and one really encounters the Father), does not thereby annul his truly mysterious character (it maintains paternal invisibility). Christ is this paradox: *his* Word protects the Father's invisibility. However, it is the Word *of the* Father to dispense salvation to all who accept it (Irenaeus, *Ad. haer.* IV, 20, 7).

This paradoxical Word, when it resorts to human logic to express the mystery of truth, resorts to the "parable". This is a broad and sapiential literary resource. "The purpose of this, however, was not to conceal its meaning, but rather [. . .] to challenge the hearer, to provoke him to attention, so as to impress the point more deeply upon him" (*Theo-Logic II*, p. 74). Thus, he brings out the mystery of glory that hides the truth announced. Jesus, "the archetypal image of God", makes the coming of the kingdom of God understandable "through images (using every form of parable)" (*Theo-Logic II*, p. 76), because he "appeals to an at least ethical and, to a large extent, also religious (pre-) understanding on the part of his hearers and that he knows perfectly well he can do so". For Balthasar, there is "more than a mere alphabet; it is a developed language, which he can take for granted and make use of for the higher transcendence that he intends". Jesus appeals "a priori to a readiness in the hearer to go beyond what he can immediately understand to the meaning Jesus intends". Indeed, human logic was prepared for divine logic. "There is a grammar here in which Jesus can encode the divine Word", albeit always illuminated by the Spirit, "who now begins in earnest to exposit for man's spirit what has already been objectively exposited in itself (Jn 16:13–15)" (*Theo-Logic II*, pp. 77–79). Thus, the relationship between parable and message is very deep and internal. The divine image (Christ) and the parabolic image are in analogical harmony. For this reason, the parable demands faith, and "only then can man attain the right action and conduct in following him", for it is a matter of understanding "God's 'praxis,'" which is the ultimate meaning of the parable, and thus becomes its norm of life (*Theo-Logic II*, p. 80). Balthasar concludes:

> Perhaps no example shows so clearly as do Jesus' practically oriented parables (along with their exigencies and consequences) how divine logic can and will express itself in human logic on the basis of an *analogia linguae* [analogy of language] and, ultimately—in spite of all objections—an *analogia entis*, fulfilled in Christ, who is God and man in one person (*Theo-Logic II*, p. 81).

Divine logic can be expressed in human logic because there is an analogia linguae as a part or expression of the analogia entis. Thus, literature, as a 'feast of language', is an integral part of *Theo-Logic*; thus, the eminently theological character of literature appears again clearly and in a new way, without this in any way detracting from its autonomy as human speech. What Balthasar discovers in Pascal applies here: Christian truth is grasped by "a conviction in its inner measuredness and its outward fitness". Literature must "to communicate wisdom with good taste (*goût bon*)". (*The Glory of the Lord III*, p. 235) This is how Balthasar constructed his trilogy using literature as one of its structural elements, which has given important novelty to his work.

## 5. Conclusions

Balthasar built his trilogy with literature as one of its structuring elements, which gives his work important novelty. His biography shows us that, at important moments of his intellectual development, poetry, theater, and the most important classical authors of all times gave him a light that, in a certain sense and at those moments, neither theology nor the established church doctrine could give him the same clarity and strength. Moreover, his doctoral training left him forever structured as a Germanist. His approach to literature was marked by the New Germanistics, which sought to understand, in a synthetic and all-encompassing way, the intellectual history that guided literary and artistic works in general.

Thus, all his subsequent theological development, without losing his own disciplinary autonomy, was carried out from that perspective that made literature a theological word, in and through its direct narrative meaning. The concepts of *form*, *style*, *Theo-drama*, *role*, *character*, *expression*, and others that we have been able to study are literary concepts applied internally to theology, and they structure Balthasar's trilogy to an important degree.

The reasons Balthasar gives for this theological use of literature are metaphysical and Christological. It is about the creative plan in Christ, the Archetype of all; the analogia entis expressed in the analogy of the transcendentals; and the concrete analogia entis that is the Word made flesh, where God speaks in and through the flesh and the human word. These three philosophical–theological principles allow for and lead to a profound *harmony* or justesse between the literary form or style, the product of the creative freedom or charisma of the writer, the content as a fragment of Being and of the divine Logos, and its openness as a voice of God from the narrative and the particularity situated in time and space. This, which Balthasar also called analogia linguae, is a "first language of God", which can be described as a "sonorous silence", a beauty that impacts and captivates, or logoi that expresses God's action in the midst of human existence and throughout the lived existence of human beings.

Accordingly, Balthasar gives literature a high theological status, which has been called a "theological a priori", a "referentiality to the *Other*", or the quality of a "theological *locus*". In this way, he incorporates literature into his theological structure, especially in his trilogy, not extrinsically—as an external example—but as an authentic word of revelation, albeit always retaining its quality as literature and not as explicit theology—though always as the speech of God. This is reflected, on the one hand, in the theological interpretation he gives to a very wide range of literary authors and works, and, on the other hand, in the use—somewhat "deconstructed" and rebuilt—of literary concepts and structures in the development of his theology.

To assess the theological contributions of this attempt, the literary justice of the method, or the real possibilities of this path is a matter for another time—although some things have been said throughout the text. Nevertheless, it was first necessary to describe the roots and the procedure of this theological use of literature, and to show how internally his Germanistic training influenced not only his intellectual formation, but above all, his theological contribution, as well as its expected limitations. I hope that I have achieved the objectives of this text.

**Funding:** This research received no external funding.

**Institutional Review Board Statement:** Not applicable.

**Informed Consent Statement:** Not applicable.

**Data Availability Statement:** No new data were created or analyzed in this study. Data sharing is not applicable to this article.

**Conflicts of Interest:** The author declares no conflict of interest.

## Notes

[1] "*Germanistik* is an interdisciplinary field concerned with the languages, literature, philosophy, religion/mythology, cultural history, and artistic productions of the broader Germanic peoples, from their tribal origins up to modern times. In light of the subject matter, the modern *Germanistik* curriculum presupposes an interconnection between all of these aspects of a given ethno-linguistic group, which is the justification for handling issues of history, culture, literature, and linguistics together" (King 2016, p. 1, n. 1).

[2] His studies and anthologies on Augustine, Irenaeus, Origen, Gregory of Nyssa, and Maximus the Confessor, which were later published, date from this period (von Balthasar 2005, pp. 162–76).

[3] "The years in Lyon were also the years in which I discovered the great French poets; Claudel, Péguy and Bernanos became life companions that I could not renounce" (von Balthasar 2007, p. 12). See (von Balthasar 1993b, pp. 88–91).

[4] See, for example, the more than 100 translations made by him, of different types of literature (von Balthasar 2005, pp. 162–76). See also (von Balthasar 1993b, pp. 13–14).

5    In addition to various references in his writings, we may mention "The farewell tercet", referring to Mozart's opera *The Magic Flute* (von Balthasar 1943), and "Recognition of Mozart" (von Balthasar 1955).

6    Augustine adds: "It is not a knowledge of little value that if before we come to understand what God is, we can already understand what he is not" (*De Trinitate*, Book VIII, chp. 2, n. 3), quoted in (von Balthasar 1984a, p. 255, note 8).

7    He quotes Ignatius of Antioch, *Letter to the Ephesians*, 19:1.

8    King explains that New Germanistics is the way Balthasar's professors approached "knowledge, history, and texts in general". "And when Balthasar came to the study of theology proper, he simply mapped that approach onto theological topics" (p. 298).

9    It is also one of Carpenter's (2015, p. 184) conclusions.

10   Especially, but not exclusively, in *The Glory of the Lord II* and *III*.

11   Bearing in mind that it is not possible to simply equate beauty with glory. This is a subject we cannot go into here, but it was an important critique of Balthasar's aesthetic project. See Brown (2018), and the literature he reviews.

12   Translated by me from the original German.

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
