# Peer review of "Harmony Justesse Analogia Linguae: Literature as a “First Language of God” in the Thought of Hans Urs von Balthasar"

_religions, doi:10.3390/rel15010113_

Round 1
Reviewer 1 Report
Comments and Suggestions for Authors
The article shows a very accurate knowledge of the profound theological insights of the Swiss Germanist Hans Urs von Balthasar, especially of the relationship between theology and literature, or more precisely of literature as a form of theology in the Christian tradition. The attempt to show how this relationship is established in three aspects (literature integrated in the theological structure; theological language in all literature; the use of some literary instruments to do theology) seems quite interesting and sustainable, especially in trying to show how the Balthasarian project has at its centre the idea of hamornia or convenience in Saint Anselm.
However, it is precisely in this element that the article is weak, since the author does little to explore Balthasar's reading of Anselm, limiting himself to saying that the Swiss author takes his central thesis from the scholastic monk, and there is nothing really new in that.
1. The author doesn't seem to problematise the consequences of reading Balthasar on the question of "convenience" in Anselm, and even less does he problematise the question itself in Anselm in order to analyse whether Balthasar overcomes some of the problems of medieval scholasticism or merely reproduces them in an aestheticised version of the problem.
In Anselm's pulchritudo rationis there is an attempt to attribute the incomprehensible divine reasons in the form of an absolute and harmonious perfection of his action in the world. Convenience is here broken down into two points: as the need for a language that is convenient for human beings, especially rational human beings, and as the convenience of God's will in its permissiveness about certain things in which divine action is appropriate, in the sense that, as Augustine says, it would have a greater advantage (a greater good) than avoiding it. In this way, expediency is part of the "doctrine of divine universal causality" present in De Concordia, Monologion, De libero arbitrio.
Such expediency unfolds politically in the reception of this argument for the necessity of evil (necessary evil) in Leibniz's theodicy and in the whole Enlightenment and even contemporary critique of forms of theodicy as divine justification for human action, especially political action, on the model of Carl Schmidt's political theology, thus invoking the divine will as the exceptional action by which God saves the order of the world. Divine convenience supports the need for the will to conform to representations of God's will.
2. In this sense, and in the way that the author limits himself to identifying the question of expediency in Anselm, he shies away from criticising the limits of Balthasar's theological aesthetics. For example, while in the theodramatic Balthasar focuses on the beauty of the Father's love in the face of the suffering of the Cross (despite the Anselmian view that the Son was sent to die for the Father himself, cf. Cur Deus homo), other theologians point to the perception of the Cross from the narrative of the Crucified, to the living of the experience of the Cross from the sense of abandonment.
Anselm tries to rescue a logic of perfection that metaphysics imputes to the question of God, and Balthasar would only present the dimension of the theodramatic beauty of the divine will.
3. In this sense, the author also fails to critically address Balthasar's position that the only figure capable of disclosing the beauty of God is the figure of Christ, without considering that the defence of this theoretical construct does not hold up in the face of historical forms of theodicy in which such beauty has served to legitimise various forms of abuse, as in the case of African colonisation through Christian indoctrination.
Of course, the Balthasarian project can be thought of against the background of Balthasar's experience of the Spiritual Exercises of Ignatius of Loyola, as a former Jesuit seeking a communion of wills between the exerciser and the discernment of God's will for his life. But this would not eliminate the task of purifying the abusive historical receptions of the divine will, in order to help the contemporary reader better perceive the depth of analogy between the Christological narrative of the search for God's will and the contemporary individual's discernment of a search for meaning and/or vocation under the sign of the search for God's will. This task requires, according to Jauss' aesthetics of reception, for example, a literature that offers a critical reading of history, so that the aesthetic experience of literature opens up new horizons for the reader. By implicitly assuming the Anselmian ratio in Balthasar, the proponent of the article runs the risk of not recognising the legitimate reasons for rejecting representations of the divine will, and runs the risk of attributing the reason for someone's non-conformity with the divine will to a lack of faith. Voltaire, for example, ironically pointed to the desperate attempt to show how the Lisbon earthquake of 1755 had a convenient purpose for the divine will, by trying to show the pulchrum of bodies split in half by the catastrophe, allegedly promoted for the conversion of the Portuguese. The Jesuit Malagrida, for illustration, who was a contemporary of the earthquake, was responsible for this effort of Anselmian convenience regarding the historical episode. This means that even if the experience of the Ignatian Exercises is a retreat from argumentation, there is still a need to purify modern theodicies.
4. In addition, the argument that all literature has a theological form is rather presumptuous, in that there is a Romantic view of what literature is. Here, too, the author either remains silent or agrees with the Swiss Germanist's assertion.
5. Still on the question of literature as locus theologicus, there is a whole discussion on the tension of the term between Felipe Melanchton and Melchior Cano about the instrumentalisation of literature as a servant of theology, since the cultural forms of expressing faith would be inferior to the theological forms (Bible, Tradition, Magisterium) and would only serve to confirm the superior forms. This discussion has been widely explored, especially in Brazilian theological research. See, for example, Manzatto, Magalhães, Barcellos and Villas Boas on the problem of literature as locus theologicus and locus revelationis.
In general, as an exegesis of Balthasar's work, which tries to grasp literature as the first theology from the point of view of linguistic analogy, there is a potentially very rich contribution. However, the author does not deal with what he has chosen as the central category, namely the Anselmian conventionality in Balthasar, and thus shows himself to be uncritical. One can consult, for example, the criticisms of Carlos Mendonça Alvarez, Renato Vieira, Jesus Martinez Gordo, Johann Metz and Alex Villas Boas.
In our opinion, there is great potential for contribution in the text, but such criticality is indispensable if we are to grasp Balthasar's precious contribution without romanticising it.
Author Response
I very much appreciate the comments from the three reviewers.
Reviewers 2 and 3 do not ask me for any corrections and largely approve of the article.
I appreciate the opinions that the article “serves a strong need for elucidation of the philosopher/theologian’s overall processes and nuances.” And also, that “the article’s author has managed to capture the essential features of this theology and present them in a clear and accessible way to readers.”
Also, reviewer 1 thinks that “the article shows a very accurate knowledge of the profound theological insights of the Swiss Germanist Hans Urs von Balthasar, especially of the relationship between theology and literature, or more precisely of literature as a form of theology in the Christian tradition.” And further that “there is great potential for contribution in the text.”
But he asks me for greater criticality, which he exemplifies in five themes.
It seems to me that, being five themes that are debated around Balthasar’s proposal, it is certainly possible to take a position on these five themes that he asks of me. Even several of his questions on Balthasar’s thought I also share them, at least in part. But I believe that they are topics that deviate from the objective of the article. That is why I have not addressed them in the text. It seemed neither necessary nor opportune to me. Each article should have only one central objective and not stray into other topics, even if they are important.
The aim of our text is “to show, in a synthetic and concrete way, how deeply Balthasar’s literary training has influenced the development of his theology.” “Our goal is to comprehensively describe the congruence that justifies its incorporation as theological data in the elaboration of Balthasar’s trilogy.” And that is why it “studies the inner harmony or convenience (Konvenienz-justesse-analogia linguae) between literature and theology as the basis for the theological use of literature.” This is explicitly stated at the beginning of the text.
Therefore, I do not agree with the substance of the criticism of reviewer 1. The reviewer assumes something that I did not say in the article. He states that I try “to show how the Balthasarian project has at its center the idea of hamornia or convenience in Saint Anselm”. And, therefore, “the author does little to explore Balthasar’s reading of Anselm, limiting himself to saying that the Swiss author takes his central thesis from the scholastic monk, and there is nothing really new in that.”
But the aim of the article has never been to study Anselm’s thesis, nor to review the concept of ‘convenience’ proper to Anselm’s theology and philosophy. I do not say that anywhere in the article. The aim of the article is to show how Balthasar uses literature as a theological referent. That is one of the contributions and novelties in the Swiss theologian, because he uses literature in a much deeper way than other theologians. A reader may agree or disagree with the results that Balthasar obtains from this use, but the objective of the article is not to criticize or not those theological results, but to show what are the reasons that Balthasar must justify this use, and how he does it. To explain this use of literature, I have used, by way of example, the word “convenience” which Balthasar takes from his explanation of Anselm, or “justesse”, taken from his explanation of Pascal, or “first language of God” or “analogia linguae”, which Balthasar uses at other times. But I do not assume the theology of each of these authors, nor do I refer to the understanding of the theology of these authors according to Balthasar, but only use in a very broad way some expression that Balthasar uses at various times and in a broad way, to explain the use of literature. Precisely for this reason I place these four expressions that come from different theological environments. I think the other reviewers understood this very well.
That is why those 4 terms appear in the title of the article and then I return to them in the conclusion, but it is understood that they are broad terms that help me to express the content of the text. The true understanding of these terms is given by the content of the article and not by the theology of Anselm, Pascal or anyone else. In fact, the reviewer 1 says nothing about Pascal's use of justesse.
Perhaps, to avoid this misunderstanding, when these words appear on p. 13, I can add a paragraph explaining that the use of these words is rather generic and does not necessarily mean that I assume in the text the theological content of the authors that Balthasar cites. And add that I am explaining the argument that Balthasar uses to use literature, not whether his interpretation of Anselm, Pascal, or some other interpretation is correct, or its consequences.
What I have just said becomes even clearer if one looks at the content of the five suggestions that reviewer 1 develops:
Suggestion 1 says that I do not make a judgment about the theodicy that can be derived from the use of the word “convenience.” This topic is outside the scope of this article.
Suggestion 2 asks me to criticize the limits of Balthasar’s theological aesthetics, which is beyond the scope of the article.
Suggestion 3 asks me to study the consequences of Balthasar’s concept of beauty, particularly in what it has meant, for example, for European colonization. Clearly, this is also beyond the topic.
Suggestion 4 asks me to take up Balthasar’s own romantic view of literature. This background of Balthasar is evident, and I say so explicitly. But to make a critique of this aspect of the author is another matter.
And suggestion 5 refers to the use of the concept of “locus theologicus”, which would imply an instrumentalization of literature as a servant of theology. That is an interesting contemporary discussion, but it also escapes the topic.
Finally, synthesizing the suggestions, reviewer 1 thinks that I am uncritical of the central category I have chosen for the article, which would be “Anselminian conventionality.” But that is not correct. I have not chosen that category. What’s more, I have used a few words (several, not only Konvenienz), among which one of them Anselm also used, but I have used them in a broad sense, which is understood throughout the reading of the article, and which can be described as a “first language of God.”
In any case, welcoming the substance of these suggestions, all this could be solved if I would add a paragraph after using these concepts for the first time, explaining that I use the terms and not the theology behind them in the authors who also use them, as it seems to me to be clearly expressed in the complete reading of the article. I think that is precisely why the other reviewers did not say the same, since they realized that it was only a use of terms and not of the theology of those authors. In short, the article shows how Balthasar uses the literature, and not whether in that use what he deduces is correct is not correct. This would be a subject for another article, and these are subjects that I have already dealt with in other articles, and even in the books I have written on Balthasar. That is why it should not be repeated here.
Finally, in the final version, I can send the text accepting the few suggestions about grammatical correctness that reviewer 2 asks for.
In summary, I propose that I send a definitive version of the article, in which, along with correcting the grammar issues indicated, I will add a paragraph explaining what I have said above about the use of the words: “convenience”, “locus theologicus”, “justesse”, “analogia linguae”, and “first language of God”.
I remain attentive to your comments and reiterate my gratitude for the very careful evaluation made by the three reviewers. They can certainly help to improve the final article.

Reviewer 2 Report
Comments and Suggestions for Authors
This article offers a strong argument for Balthasar's creation of a literary theology. There are numerous fine points made and it seems to insert a clarity of Bathalsar's purpose within his larger treatises. I believe it serves a strong need for elucidation of the philosopher/theologian's overall processes and nuances.

The article is well-written; I have made only a few suggestions about clarification and/or grammar.
Author Response
I have accepted all the suggestions about clarification and/or grammar that reviewer 2 has pointed out to me.
In the revised version I have highlighted them with yellow.
Kind regards,
Reviewer 3 Report
Comments and Suggestions for Authors
Hans Urs von Balthasar has written a theology of great depth that is a challenging read. The article's author has managed to capture the essential features of this theology and present them in a clear and accessible way to readers. Balthasar's generous vision of the theological essence of art, literature, and drama is summarized and explained in this well-written and amply documented article.
Author Response
Thank you very much for the kind comments of reviewer 3.
Reviewer 3 does not request any changes.
Round 2
Reviewer 1 Report
Comments and Suggestions for Authors
Dear Author,
Thank you for taking the time to respond carefully to my comments. In fact, we don't have to agree on everything to acknowledge the quality of the work I'm affirming here.
These are opportunities for good dialogue. And that's why I come back to your comments.
Actually, I didn't make suggestions, but rather critical remarks from the point of view of the reception of Balthasar's work, and these remarks were aimed at identifying why such issues were absent from this article. These questions are part of the reason why Balthasar's work has not had such an impact outside the field of theology, as Venard, quoted in the text, points out.
The reason for my first observation about Anselm's technical term is that it is expressed in the title of the first version, as well as in the abstract of his statement: "Balthasar bases this profound and internal use of literature on convenience". I agree that "harmony" less compromises the aim of the text and opens up the possibility of thinking about a critical analysis of Anselm's reception in Balthasar and the extent to which they agree. For the purpose of the text, the notion of harmony may be sufficient.
The second reason for my observations lies in the statement in the first version:
"However, none of them explicitly studies the inner harmony or convenience (Konvenienz-justesse- analogia linguae) between literature and theology as the basis for the theological use of literature: what we have called the theological voice of literature-although they indirectly alluded its reference. Our goal is to comprehensively describe the congruence that justifies its incorporation as theological data in the elaboration of Balthasar's trilogy, the culminating work of this great author."
In this case, the statement would have three problems, namely
1) the possibility that other authors (in addition to those quoted) have not dealt with the problem of harmony and its political consequences in forms of theodicy, referring to notes 1, 2 (criticism by Rahner, for example), 3 (criticism by some of the authors quoted, including myself, in articles and books). As I said, the notes are not suggestions for inclusion, but aim to understand why such problems are absent; although they are not the scope of the article, the fact that something has been said in relation to the Anselmian problem raises the question of whether such questions are implicit in the author's perspective. Replacing the question of convenience with harmony might be a way out of attacking the problem in this article.
2) If the aim of the article was to "comprehensively describe the congruence" between Konvenienz [Anselm's], justesse [Pascal's] and Balthasar's analogia linguae as a theological basis for thinking about inner harmony, this would inevitably and inescapably unfold in the problems of theodicy, and a good number of 19th and 20th century literary authors not cited by Balthasar would point to it as a reason for rejecting God.By adopting the concept of 'harmony', one might suggest that the author is focusing on form or style rather than on the theological problem involved. In this sense, I would venture to say that harmony is more a tributary to Pascal's justesse and its correlative spirit of finesse. And Anselm is seen here as a link in the tradition. In this sense, harmony can be seen more as a "generous symphony", as Christopher Denny has put it, a generosity that allows the Swiss to accept Anselm uncritically. In this sense, the addition made in lines 436-439 is fundamental: "the voice of God is in the concrete story, but it is not exactly the story. It is necessary to know how to find behind the story the mysterious Word of God, spoken in a language that only the "heart" understands, because it is truly an encounter with God”.
3) In the following statement in lines 69-70, it is not clear to me whether the author of the article understands the question of harmony to be "the" [only form of] "basis for the theological use of literature", or whether he/she is referring to Balthasar's basis for the dialogue between theology and literature. Again, if it is the author's statement about "the" basis, this opens up a discussion of major proportions; if it is the statement about Balthasar's basis, it would be interesting to make it more explicit.
If you could also clarify the statements that "all literature has a theological form" and "the basis for the theological use of literature" (whether yours or referring to Balthasar's statements), it would be very timely.
In short, I hope that the author understands that my observations are the result of a careful reading and that he has appreciated the work presented, which I like very much, and that I hope to read the author once the text is published and I know the authorship. My remarks here are not intended to diminish the work, but rather to help polish the edges of a very lively debate in the interdisciplinary dialogue between theology and literature, which is very precious to me. Congratulations on an excellent contribution!
Author Response
Dear reviewer,
Thank you very much for your very wise suggestions, both in the first review and in this last one. I have taken very much into account everything you have told me, and I believe that the article has been improved thanks to your comments. I will be glad to get in touch with you once everything is published, to exchange views.
Regarding the comments of this second revision, I can say the following:
- I have followed your suggestions to avoid the concept of Konvenienz, to use instead the more general concept of harmony, which "less compromises the aim of the text and opens up the possibility of thinking about a critical analysis of Anselm's reception in Balthasar and the extent to which they agree."
- With respect to the three problems with the quoted paragraph, I have followed your advice.
a. Replacing the question of Konvenienz with harmony.
b. Indeed, I think that the concept of harmony is much more in line with Pascal's justesse, and does not compromise Balthasar's interpretation of Anselm. That is why, in this final version, I have left the concept of justesse, but have eliminated that of Konvenienz, except once, where I place it in Anselmo's mouth and without compromising the interpretation of our theme.
c. I have clarified the somewhat ambiguous statement in lines 69-70. It is not the basis, but simply a basis (or hermeneutical criterion) for the theological use of literature, and there may be many other bases for the theological use of literature, both in Balthasar and in other authors. Although Balthasar, from his aesthetic perspective, privileges this one. I have tried to make that more explicit in the text.
I have also clarified the two sentences that raised some doubt in their interpretation. Instead of speaking of all literature, in several of the passages I spoke rather of literature in general, in order to avoid a too strong statement, as well as to make some sentences more explicit. And I added that the harmony of literature and theology is a basis for the theological use of literature (there being others); and that refers to Balthasar's thought.
I reiterate my gratitude to the reviewer, and I hope with this I have clarified all the doubts, so that the readers can know something more of the thought of this author who always provokes many reflections.
Sincerely yours,